# Environmental Impact of Rotationally Grazed Pastures at Different Management Intensities in South Africa

**DOI:** 10.3390/ani11051214

**Published:** 2021-04-22

**Authors:** Hendrik P. J. Smit, Thorsten Reinsch, Pieter A. Swanepoel, Ralf Loges, Christof Kluß, Friedhelm Taube

**Affiliations:** 1Institute of Crop Science and Plant Breeding, Grass and Forage Science/Organic Agriculture, Christian-Albrechts-University Kiel, D-24118 Kiel, Germany; treinsch@gfo.uni-kiel.de (T.R.); rloges@gfo.uni-kiel.de (R.L.); ckluss@gfo.uni-kiel.de (C.K.); ftaube@gfo.uni-kiel.de (F.T.); 2Department of Agronomy, Stellenbosch University, Stellenbosch 7600, South Africa; pieterswanepoel@sun.ac.za; 3Grass Based Dairy Systems, Animal Production Systems Group, Wageningen University (WUR), 6700 HB Wageningen, The Netherlands

**Keywords:** farm-N-balance, carbon footprint, dairy, greenhouse gas, sustainable agriculture

## Abstract

**Simple Summary:**

Nitrogen fertilization, irrigation and concentrate feeding are key management factors in grazed dairy-pasture systems. However, the extent to which these management factors affect environmental efficiency is a current debate among scientists. We designed a study to investigate dairy system environmental efficiency through the evaluation of the carbon footprint of milk and the nitrogen-balance as a result of different fertilization rates applied on irrigated dairy-pastures in South Africa. The lowest carbon footprint was observed when low rates of fertilizer were applied. Methane from ruminal digestion and nitrous oxide emissions from grazed pastures were the main contributors affecting the carbon footprint. The application of fertilizer resulted in only small herbage yield differences between treatments. The nitrogen-balance was negative when no nitrogen was applied. This indicates that such an approach will not be sustainable without adapting sward species composition (e.g., introduction of forage legumes), even though nitrogen circulates through animal manure to the pasture. The environmental impact of milk produced from pasture-based dairy farms can be reduced through increased farm nitrogen use-efficiency and improved irrigation systems in South Africa.

**Abstract:**

Nitrogen fertilization, irrigation and concentrate feeding are important factors in rotational pasture management for dairy farms in South Africa. The extent to which these factors affect environmental efficiency is subject to current and intense debate among scientists. A three-year field study was conducted to investigate the yield response of different N-fertilizer treatments (0 (N0), 220 (N20), 440 (N40), 660 (N60) and 880 (N80) kg N ha^−1^ year^−1^) on grazed pastures and to calculate the carbon footprint (CF) of milk produced. Excessive N-fertilization (N60 and N80) did not increase herbage dry matter and energy yields from pastures. However, N80 indicated the highest N-yield but at the same time also the highest N surpluses at field level. A maximum fertilizer rate of 220 kg ha^−1^ year^−1^ (in addition to excreted N from grazing animals) appears sufficient to ensure adequate herbage yields (~20 t DM ha^−1^ year^−1^) with a slightly positive field-N-balance. This amount will prevent the depletion of soil C and N, with low N losses to the environment, where adequate milk yields of ~17 t ECM ha^−1^ with a low CF (~1.3 kg CO_2_ kg ECM^−1^) are reached. Methane from enteric fermentation (~49% ± 3.3) and N_2_O (~16% ± 3.2) emissions from irrigated pastures were the main contributors to the CF. A further CF reduction can be achieved by improved N-fertilization planning, low emission irrigation techniques and strategies to limit N_2_O emissions from pasture soils in South Africa.

## 1. Introduction

In South Africa there is a rising trend for dairy production based on grazed pastures. This is mainly due to production based on pasture utilization being more cost effective than confined or semi-confined systems that use “bought in” forage. Despite the declining number of milk producers in South Africa, the total milk production has increased by 26% in recent years (2011–2019) [1]. Most dairy farming systems in South Africa are based on pastures providing most of the cow’s nutritional requirements [2]. The majority of milk production is in the Western Cape, Eastern Cape and KwaZulu-Natal Provinces [1]. Dairy cows are allowed to graze pastures year-round without any confinement. South African pasture-based systems are not representative of sub-Saharan Africa, but are more comparable to systems in New Zealand and Australia, as the same management strategies are adapted and followed. The botanical composition of dairy pastures consists mainly of a kikuyu (*Pennisetum clandestinum*)-base that is over-sown with ryegrass (*Lolium* spp.) during autumn to ensure fodder production throughout the year [3]. However, supplementation in the form of grain-based concentrates are fed in the milking parlour to address possible nutrient deficits and to ensure a high milk yield [4]. It has become a common practice in pasture-based systems to feed a concentrate mixture during or after milking. Concentrate feeding may replace part of high-quality forages, such as cultivated pastures [5]. This can increase the stocking rate but as a result it also increases the risk of excess nutrients circulating within, or being lost, from pasture-based systems.

The performance of pasture-based dairy farms is dependent on herbage yield and its quality. Irrigation and fertilizer can increase herbage yield of pastures and thereby enable increased stocking rate and milk production [6]. Availability of water is usually a limiting factor due to insufficient precipitation, and therefore pastures in some areas are additionally irrigated. Dairy farmers generally use high amounts of fertilizer to promote herbage growth and maximize herbage yield per hectare, despite the high amounts of manure that is available. In 2015, South Africa consumed around 427,000 tons of nitrogen fertilizer [7]. However, there is often no additional herbage yield response obtained from high amounts of applied nitrogen relative to moderate inputs [8,9]. The field N-surplus can exceed >400 kg N ha^−1^ year^−1^, particularly if N is applied in high rates [9]. The risk of nitrate leaching and water pollution is increased if N is applied in excess of plant needs [10]. Moreover, the excess of N leads to elevated N_2_O emissions from managed soils [11,12,13], which has already been demonstrated for irrigated pasture systems in South Africa [9]. N_2_O is seen as a potent GHG which has a 121-year atmospheric life span and a global warming potential of 265 times that of CO_2_ compared over a 100-year period [14]. This makes abatement strategies of N_2_O critical to control GHG emitted from managed soil in the agriculture sector.

Tools are needed to assist in the development of policy strategies which allow the dairy sector to thrive while minimizing GHG emissions [15]. The growing concern over GHG and the effect of dairy production on the environment has led to the need to express the total emissions associated with milk. Product carbon footprint (CF) analysis has become broadly accepted as a method to assess the impact of a product in relation to its effect on the environment. The CF of dairy production systems can be calculated and expressed as the net exchange of all GHG in CO_2_ equivalent units per unit of energy-corrected milk [16] produced. The N-fertilizer management is an important parameter to consider when calculating the CF of milk [17] as increasing levels of N applied are associated with elevated N_2_O emissions from irrigated pasture soils [9,18,19]. Currently in South Africa, only a few attempts have been made towards calculating the CF of milk from pasture-based systems. These studies suggest that the CF of milk ranges from 0.94 to 2.07 at the farm gate, but little is known on the impacts that N fertilizer management has on the CF of milk in South African dairy systems [6,18,19].

Against this background, the aims of the study here are to evaluate dairy-pasture systems in terms of different N fertilization levels on herbage yields as well as calculating the associated CF of milk. The study endeavors to fill this gap in the literature while contributing to improved knowledge on sustainability of milk production by assessing the CF of milk produced at different management intensities in pasture-based systems. Accordingly, in this paper, we present results from field trials over a three-year period to address the following questions: to what extent does mineral N-fertilizer usage in intensive rotationally stocked dairy pasture systems in South Africa affect the milk yield, CF, N-balance and N-footprint per hectare and per kg of milk produced?

## 2. Materials and Methods

### 2.1. Experimental Site Description

A field experiment was conducted at Outeniqua Research Farm (33°58′38″ S; 22°25′16″ E; 201 m.a.s.l.) of the Western Cape Department of Agriculture, South Africa. The research farm is located near the city of George in the southern Cape region of South Africa. The area has a temperate climate with mean monthly temperatures ranges of 7–18 °C and 15–25 °C in winter and summer, respectively. Rainfall is evenly distributed throughout the year and has a mean annual average precipitation of 661 mm (ten-year data). The soil type on the experimental site can be classified as a Podzol [20] which is locally known as a Witfontein soil form [21]. Winter was defined as months June–August, spring as September–November, summer as December–February, and autumn as March–May.

### 2.2. Pasture and Grazing Management

The experimental site consisted of kikuyu over-sown with ryegrass and managed under no-tillage practices. Prior to 2016, the pastures were uniformly fertilized at ca. 40 kg N ha^−1^ after every grazing period, for multiple years. During over-sowing of ryegrass, pastures were first grazed to a height of ca. 5 cm above ground level and mulched afterwards. In autumn 2017, 2018 and 2019 the kikuyu was over-sown with annual Italian ryegrass (*Lolium multiflorum* ssp. *italicum*). The over-sowing seed rate was 25 kg ha^−1^.

Pastures were managed under permanent sprinkler irrigation with 15 m spacing between sprinklers. Irrometer tensiometers (Calafrica SA, Nelspruit, South Africa) were installed at a depth of 15 cm and irrigation was scheduled to maintain a soil water matrix potential between −25 and −10 kPa. There was 365-day access to pastures as dairy cows are kept outside year-round. The pastures were intensively grazed by Jersey cows with grazing cycles of between ca. 28 days in summer and ca. 35 days in winter. Cows were allowed to voluntarily graze the experimental plots and were not restricted to specific treatments. The cows grazed the swards to ca. 5 cm above ground level so that residual effects in determining herbage yield were considered minimal.

### 2.3. Experimental Layout of the Pasture Experiment

An experiment was laid out as a randomized block design with five N fertilizer rates as treatments, replicated in four blocks. Plots were 15 × 15 m. The study was conducted over a three-year period from April 2016 to June 2019. The years 2016/2017, 2017/2018 and 2018/2019 are hereafter referred to as year 1 to year 3, respectively. The prevailing dairy production parameters considered (average showed in brackets over the three years) were as follows: grazing cycle (12 cycles), farm area (121 ha), number of dairy cows (454 cows), breed (Jersey), live weight (380 kg), replacement rate (25%), stocking rate (4.7 livestock units ha^−1^), calving interval (365 days), energy corrected milk (ECM; 16.7 t ha^−1^ year^−1^), milk fat (4.9%) and milk protein (3.7%). Cows were allocated between 8 and 10 kg DM pasture with an additional concentrate feeding in the milking parlor of 5–6 kg concentrate. Nitrogen fertilizer was applied by hand in the form limestone ammonium nitrate (LAN) at five fixed rates. The five rates consisted of a control (no N) and 20, 40, 60 and 80 kg N ha^−1^ (respectively, defined as N0, N20, N40, N60 and N80) applied after every grazing event. Twelve split dressings of fertilizer were applied per year. The average annual N application rates for N20–N80 were 220, 440, 660 and 880 kg N ha^−1^ year^−1^ during the three experimental years.

### 2.4. Herbage Yield and Forage Quality

Herbage yield was sampled by cutting the pasture sward prior to each grazing period. Metal rings with a diameter of 35.6 cm were used to take biomass samples at a height of 3 cm above soil level. Ten ring-samples per plot were cut by hand shears and collected in a bag. The rings were placed randomly within the different plots. Samples were placed in an oven (ODS, 1400 L, SMC Oven, Killarney Gardens, South Africa) to dry at 60 °C for 72 h. The dry matter (DM) content and herbage yield (t DM ha^−1^) were then determined. Prior to analysis, dried pasture herbage samples from each plot were dried and milled (SMC Hammer mill, 1 mm sieve). Forage quality parameters were estimated using near infrared reflectance spectroscopy (NIRS). All milled samples were scanned in duplicate using a NIR-System 5000 monochromator (FOSS, Silver Spring, MD, USA). The metabolizable energy content (ME) and the net energy lactation content (NEL) of herbage was calculated according to GfE (2008) [22] and Weißbach et al. (1996) [23]. Milk yield was calculated according to the herbage yield. The total crude protein (CP) was calculated by multiplying the respective N-content with a factor of 6.25.

### 2.5. Carbon Footprint Calculation of the Different Pasture Treatments

The different treatments in this plot experiment were grazed evenly by the dairy herd of the experimental farm. Therefore, individual animal measurements for milk yield for the different fertilizer treatments were not possible. Consequently, the potential pasture milk yield of the different treatments was calculated according to the measured metabolizable net energy (MJ NEL) yield from herbage mass. The NEL requirements to produce one liter of milk was calculated according to Gruber et al. 2014 [24] using the following equation:NEL (MJ kg ECM^−1^) = 0.38 × fat (%) + 0.21 × protein (%) + 1.05(1)
where fat (4.9%) and protein (3.7%) represent the percentage of milk produced. In order to calculate the CF of the different pasture milk yields, it was assumed that requirements for maintenance energy were covered by feeding supplements at a rate of 3.2 kg DM day^−1^. The concentrates comprised a mixture of maize (50%), hominy chop (17.5%), wheat bran (9.52%), soybean hulls (9%), soybean oilcake (6.25%), molasses (4%), feed lime (2.45%), mono-calcium phosphate (0.2%), salt (0.5%), MgO (0.25%) and a premix (0.33%).

GHG emissions from purchased supplements and fertilizers as well as farm management activities (i.e., pasture management and energy requirements in the milking parlor) were taken from the ecoinvent database (vers. 3.3.) [25] (Table 1). Supplementary feed imports were considered to be from the global market, the main components of which were assumed to be from South- and North America (maize) and northwest Europe (wheat). Irrigation was distributed equally over the different treatments and was considered in accordance with local irrigation guidelines for pastures of mixed grass species [26]. The proposed amount of irrigation was set as 4200 m^3^ ha^−1^ year^−1^.

The N_2_O emissions from pastures were calculated as exponential function of the calculated field-N-balance according to Smit et al. (2020) [9], using the following equation:N_2_O emissions (kg ha^−1^) = 1.99 + 1.39 × exp(0.00488 × x (kg N ha^−1^)),(2)
where *x* is the calculated field-N-balance in kg N ha^−1^. The field-N-balance for the different treatments were calculated as a result of N-fertilizer inputs, N-excreta returns and deduction of pasture herbage N-yields. N-excreta returned during grazing was calculated by the measured pasture N intake after deduction of the calculated pasture milk-N-yield. An annual proportion of 0.83 of grazing days per year were considered throughout the CF calculations. For manure management the daily N-excretion (N_ex_) was estimated by daily DM intake, its CP content and live weight of the dairy cows, following the approach of Nennich et al. (2005) [27]. The average daily DM intake of feed by cows was estimated according to the equation from Gruber et al. (2004) [28], whereas differences of DM intake between the different treatments occurred due to the different potential milk yields from offered herbage in the prevailing pasture treatments. Indirect sources of GHG emissions as NH_3_ volatilization from N_ex_ during grazing were based on a review analysis of Sommer et al. (2019) [29], where a factor of 0.65 for total ammonium N (TAN) was used. Additional N-leaching losses were calcualted according to the IPCC 2019 guidelines [30]. Methane emissions from ruminal fermentation was calculated according to Schils et al. (2007) [31].

In order to account for the on-farm soil organic carbon (SOC) changes of the tested production systems a simple approach developed by Petersen et al. (2013) [32] was used. In this approach the different crops and management systems are compared to a reference system to estimate potential SOC changes. Carbon inputs from pasture-derived roots and exudates were calculated as a fraction of above ground biomass according to Taghizadeh-Toosi et al. (2014) [33].

The global warming potential (GWP) per hectare was calculated using the respective value for each trace gas (CO_2_ = 1, N_2_O = 265, CH_4_ = 28) over a life-span of 100 years [14] and expressed as CO_2_eq. The efficiency of the different N fertilization strategies, in relation to climate change, was calculated on the basis of the functional unit ECM as proposed by Sjaunja et al. (1990) [16].

The farm-N-balance was calculated using a simple equation which deducts the nitrogen outputs at farm gate from the sum of the nitrogen inputs. The amount of mineral fertilizer N and supplement imports accounted for N-input. The quantities of purchased mineral fertilizers was based on the experimental layout (Section 2.3).

Milk was considered as nitrogen output from the different treatments (for milk productivity see Section 2.4). The N content exported as milk was calculated by dividing the protein yield by 6.38 [34]. A total N export of meat was estimated as 8 kg N ha^−1^ on average for the experimental years over all treatments.

### 2.6. Statistical Analyses

The statistical software R 4.0.2 (2020) [35] was used to analyse the data using the packages “nlme” [36] and “multcomp” [37]. The data evaluation started by defining an appropriate statistical mixed model [38,39]. Data distribution was visually assumed to be normal and heteroscedastic with regard to the fertilization treatments. These assumptions were based on a graphical residual analysis. The statistical model included year and fertilization treatment and their interaction as fixed factor. For the yearly LCA value measurements, the year was set as random in the model. Block was regarded as the random factor. Based on this model an analysis of variance (ANOVA) was conducted to test the hypothesis of the experiment. Furthermore, multiple contrast tests (e.g., see Bretz et al. 2011 [40]) were implemented in order to compare the several levels of the tested fertilization treatments. In addition, simple regression models were developed to investigate the pair-wise dependencies of the variables milk yield, GHG emissions, CF and N-balance. Statistical significance of the tested factors, comparisons of means were considered when *p* < 0.05.

## 3. Results

### 3.1. Herbage Productivity

The response in herbage production to different levels of mineral fertilizer is shown in Table 2. The average herbage yields for the N0, N20, N40, N60 and N80 treatments during the three experimental years were 18.5, 20.2, 20.1, 20.9 and 21.5 t DM ha^−1^, respectively. Differences in the total dry matter yield between treatments were low and were significant only for N20 and N60 in the second experimental year.

Nitrogen yield per hectare showed an increase with increasing N application rates. Significant differences (*p* < 0.05) were observed between the three years as well as between treatments (Table 2). On average over the three experimental years, the pasture herbage N-yields were 537, 580, 640, 738 and 833 kg N ha^−1^ in the N0, N20, N40, N60 and N80 treatments, respectively.

Energy content (GJ NEL ha^−1^) only differed (*p* < 0.05) in year 2 from year 1, but not from year 3, for the N20 and N60 treatments. Differences within the treatments were observed in year 1 for the N0 treatment which differed (*p* < 0.05) from the N60 and N80 treatments. The only difference (*p* < 0.05) observed in years 2 and 3 was between the N0 and N80 treatments.

### 3.2. Carbon Footprint of Milk

Methane emissions resulting from ruminal enteric fermentation were on average the largest contributor (49%) to the total GWP per hectare over all treatments (Appendix A, Table A1, Figure 1). When the CH_4_ emissions were expressed as the total GWP per hectare, the CH_4_ emissions accounted for 65, 58, 50, 41 and 30% for the 0N, 20N, 40N, 60N and 80N treatments, respectively. Herbage production as a result of pasture management (Appendix A, Table A1) and direct N_2_O emissions from fertilization accounted for 28% of the total GWP. More than half of emissions in the N60 and N80 treatments were the result of direct N_2_O emissions from mineral fertilizer applied to pastures and as a result of irrigation. Mineral N fertilizers as inputs accounted for the third largest contributor (12%) of total GWP per hectare. Manure storage plays less of an important role in pasture-based systems where grazing takes place year-round and was found to account for 2% of the total GWP per hectare over the various treatments. Soil carbon sequestration had a positive effect in reducing the total GWP per hectare over all treatments (Appendix A, Table A1). Higher sequestration rates were observed in the treatments receiving more mineral-N fertilizer (N60 and N80) as a result of slightly higher biomass and associated plant residues. Calculated annual sequestration was 0.85, 0.85, 0.87, 0.89 and 0.91 t C ha^−1^ across the treatments. The CF expressed as ECM ha^−1^ increased as the ECM per treatment also increased; however, it did not differ between the N0 and the N20 treatment (1.3 kg CO_2_eq kg ECM^−1^).

### 3.3. Field Level and Farm-N-Balance

The field-N-balance, based on the N in milk yield from pasture and N-returned through animal excreta after the deduction of ammonia volatilization, revealed a range of −119, +86, +299, +501 and +706 kg N ha^−1^ on pastureland for the N0, N20, N40, N60 and N80 treatments on average over the three experimental years. However, in order to calculate the farm-N-balance, gaseous N-losses were not deducted and therefore the N-losses (Appendix A, Table A2) increased from 31 to 899 kg N ha^−1^ year^−1^. The mineral-N fertilizer accounted for the highest input of N at farm gate, followed by supplementary feed imports. Milk accounted for the largest proportion of exported N at the farm gate.

Calculated potential milk yields on the farm, for treatments based on the amount of pasture herbage on offer per hectare, were 14.8, 16.6, 16.5, 17.5 and 18.0 t ECM ha^−1^ for the 0N, 20N, 40N, 60N and 80N, respectively (Table 3). To achieve this, the land requirement needs to be 1.1, 1.0, 1.0, 0.9 and 0.9 m^2^ per liter ECM for the various treatments 0N, 20N, 40N, 60N and 80N. With increasing farm-N-balance per hectare, the GHG emissions were generally higher. This relationship could best be described as exponential (Figure 2a). In contrast, the ECM ha^−1^ correlated linearly with the farm-N-balance (Figure 2b). However, the lowest farm-N-balance (N0 treatment) did not result in the lowest CF (Figure 2c) and were similar between the N0 and N20 treatment.

## 4. Discussion

### 4.1. Herbage Productivity

The performance of pasture-based systems is critically dependent on the quality and yield of the pasture herbage. The climatic conditions in the southern Cape region of South Africa are favorable for grassland and allow high herbage yields, with values between 13.5 and 22.9 t DM ha^−1^ year^−1^ reported [3,8,41,42]. This range is in accordance with the amounts of herbage produced from pasture swards over the three-year period of this study, which was in the range of 17.4–22.9 t DM ha^−1^ year^−1^. Mineral fertilizer applied had little effect on herbage production as high yields were also observed in the N0 treatment (Table 2). As described in Smit et al. (2020) [9], this could be due to within-field N cycling and high N returned to pastures by grazing animals, which provides the majority of N in the system. Taking N-losses from ammonia volatilization into account, which have been found to be between 5 and 20% in different studies [29,43], pasture systems play a role in mitigating N volatilization, and thus have the potential to improve on-farm N-cycling compared to confinement systems [44]. Moreover, pasture herbage at our study site is typically supplemented with concentrates of 4–6 kg cow^−1^ day^−1^. This protein-rich feed provides additional N-input to the system, and thereby makes a major contribution to the demand of the control swards likely promoting adequate yields.

The highest N yield in the current study was associated with the N80 treatment. This level of N input is high by international standards of fertilizer use on grazed grassland, but local advisory information in the past has advocated high N use and there is evidence that this occurs in practice [10,45]. Excessive use of mineral N fertilizer increased the N contents in the pasture herbage significantly, thereby reaching CP contents of 250 g per kg DM herbage intake. These values are considerable and in excess of the requirement of dairy cows [46]. Moreover, it is possible that a lower proportion of intake N will end up in milk and therefore excess N will be excreted due to the metabolic cost of urea synthesis [47].

The energy contents (NEL values) were comparable among treatments except between the treatment that received the highest amount of fertilizer compared with the control treatment. As expected, the treatments receiving lower amounts of fertilizer had lower herbage CP-contents compared to the high fertilizer treatment. A decrease in CP-content will be compensated by a higher amount of water-soluble carbohydrates [48] which could explain why the treatments receiving less fertilizer had a comparable NEL-content. Furthermore, the different grass species used in the current study have different seasonal growth patterns. Metabolizable energy in kikuyu can range between 8.13 and 9.9 MJ kg^−1^ DM and the species exhibits a definite seasonal pattern [49], i.e., a low ME during late-summer and autumn [42]. Ryegrass is over-sown into the kikuyu-base in autumn to overcome this problem and to ensure a continuous supply of forage for grazing [49]. This ensures adequate amounts of energy for dairy production throughout the year and could explain the small differences in NEL-content observed over the course of the study.

Oversupply of N through fertilization can be detrimental to bacteria involved in mineralization of N from soil organic matter [50]. Sustained plant production is dependent on the potential of the soil to supply plant-available N through effective N cycling. If the soil’s ability to mineralize soil organic N is compromised, this may require greater reliance on mineral-N fertilization, the cost of which may have unfavorable economic consequences. Despite fertilization guidelines in the past recommending the use of high N inputs (300–500 kg N ha^−1^ year^−1^) on kikuyu-ryegrass pastures [10,45], the results of the current study demonstrated that N applications of ≤220 kg N ha^−1^ year^−1^ are adequate to ensure maximum herbage production (≈20 t DM ha^−1^ year^−1^) under grazing management, together with an acceptable nutritional composition for optimum dairy production. It is, however, necessary to determine the effect of fertilization of rates as low as 220 kg N ha^−1^ year^−1^ on the long-term productivity of pastures. It is also important to explore combinations of different resources (e.g., breed, feed additives, irrigation methods) to ensure that milk production is consistent with environmental goals [51].

### 4.2. Carbon Footprint of Milk

It was previously thought that the CF of milk is mainly negatively correlated with increasing quantities of milk yield. The calculated CF, from the investigated low-cost grazing system reported here, ranged between 1.3 and 2.6 kg CO_2_eq kg ECM^−1^ for South Africa (Table 3). These values are higher than studies by Galloway et al. (2018) [6] and Keller et al. (2017) [51], which reported CF values in the range 0.94–2.07 kg CO_2_eq kg ECM^−1^ and 1.2–2.0 kg CO_2_eq kg raw milk, respectively, from pasture-based dairy systems. However, the CF of the N20 treatment was within this range (1.3 kg CO_2_eq kg ECM^−1^), and this is more representative of the normal management regime and fertilizer guidelines currently followed. The high CF for the other treatments is most likely to be a result of the high amounts of mineral fertilizer applied. Moreover, regionally specific N_2_O emission factor values [9] based on the field-N-balance were used to calculate the CF of milk as suggested by various authors [13,52,53]. The relationship between field-N-balance and N_2_O emissions was exponential in our approach. In comparison, other studies considered a linear function based on the amounts of N-applied.

Liu et al. (2006) [54] reported that large fluxes of N_2_O coincided with irrigation events. In addition to the direct GHG emissions from pastures, land irrigation requires additional energy expenditure. Considering these requirements and in addition to the external mineral fertilizer use, herbage production contributes ~28% to the CF. Accordingly, Oliveira et al. (2020) [52] indicated less favorable results for the CF footprint under irrigated pastures with high stocking rates in Brazil. This highlights the importance of adjusted fertilizer planning and improved irrigation management for intensively managed, rotational pastures within the context of GHG mitigation strategies in the dairy sector in South Africa.

Methane emissions, as a result of enteric fermentation from dairy cows, ranged between 30% and 65% and comprised the highest contributor to the total GWP in the current study. The N0 and N20 treatment values are comparable to other literature values, which were between 58 and 65% of the total CO_2_eq in the CF of milk in dairy systems [6,53,55,56,57], but were found to be lower for the N60 and N80 treatments. However, most of these values were obtained from intensive confinement systems in which cows had a higher daily DM intake and consequently higher CH_4_ emissions per cow and year. The calculated DM intake of the Jersey cows used in this study was 14–15 kg DM day^−1^ and is lower than for, e.g., Holstein Friesian breeds used for grazing (19 kg DM day^−1^) or confinement systems (23 kg DM day^−1^) [47]. The increasing dominance of GHG emissions from pastures in response to increasing fertilization explains the lower contribution of CH_4_ towards the total GWP compared to other studies. Moreover, improved grazing management offering highly digestible forage ensures low CH_4_ emissions per kg DM intake [58]. Even though this effect was not captured by the formula used for CH_4_ prediction in our study, it still highlights that there is further mitigation potential if adequate and management-specific equations are used for CF calculations.

Pasture-based systems have an additional advantage in terms of GHG balance through their potential to sequester C into the soil [59]. The annual sequestration in the current study was calculated at ~850 (N0) to 1140 (N80) kg C ha^−1^ year^−1^. These values are plausible, and although at the upper part of the scale for recently seeded grassland swards in northwest Europe [60] they are lower than calculated by Galloway 2020 [61] for South African pasture-based systems. This could indicate a possible underestimation of the mitigation potential of these pastures, and therefore the potential to further reduce the CF. The higher sequestration rates in the fertilized treatments are a result of slightly higher herbage yields and associated plant residues remaining on the field after grazing. Moreover, taking only minor differences of botanical composition between the treatments into account, several studies found higher belowground productivity when fertilization and irrigation were applied [62]. In the current study, C sequestration led to an average reduction of 14% of the final CF over all treatments. This allows pasture-based systems to be considered as more favorable, in terms of environmental sustainability of milk production, than confinement systems, which are often based on arable forage crops such as maize silage that can result in a depletion of soil organic C [60].

The calculated CF in this study appears to be slightly higher than values from other temperate climate zones [63,64]. A meta-analysis of 41 dairy farms in Australia conducted by Christie et al. (2012) [65] found a range of 1.0–1.6 kg CO_2_eq kg^−1^ FCM, where 95% of the variation could be explained by prevailing grazing management practices [65]. Thus, when competing with the world market for low-emission food sources, more GHG mitigation measures must be set in place and adopted by the grazing systems in South Africa. However, according to the FAO (2010) [66], estimated emissions from Sub Saharan Africa are 7.5 kg CO_2_eq kg^−1^ FCM, which are much higher than results from our study. A possible explanation is that South African pasture-based systems are not representative of sub-Saharan Africa, but are more typical of the systems of New Zealand and Australia, as the similar management strategies are followed.

The CF of milk may be affected further by the use of supplementary concentrate feed as this is often brought in from elsewhere and it is important to include its carbon emission impact. The current study considered a low-input dairy system for the CF calculation without excessive use of supplements, but high enough to fulfil maintenance energy requirements of the cows. However, a minimum level of concentrate feeding is necessary to achieve an ideal energy to protein ratio. This is particularly true under high fertilization rates, with consequent high protein contents in the herbage, as was the case here in the N80 treatment. However, the energy yield from pastures did not differ significantly between treatments, according to the field measurements. It is therefore likely that a higher demand for energy-rich supplements is necessary in the N80 treatment to reach similar energy/protein ratios across all fertilizer treatments. As a result, the GHG emissions in the N80 treatments may be slightly underestimated and require an additional amount to include the CF of imported supplements. However, the focus of the current study was on herbage provided from pastures in relation to different fertilization rates, rather than that of concentrate production elsewhere.

### 4.3. Farm-N-Balance

In the current study, the main source of N came from purchased mineral fertilizers. Consequently, the fertilizer and the extensively imported supplements were the most prominent factors influencing the field- and farm-N-balance, respectively, as well as the N-footprint for milk. In South African pasture-based dairy systems, cows graze pastures year-round, which makes high N returns to pastures more likely. Dairy cows excrete ~75% of their N intake, whereas less than ~25% is metabolized into the milk output from the pasture system [67]. Even if concentrates are fed at low levels and ammonia volatilization after excretion of urine-N is considered, there will still be a considerable amount of N returned through excreta if high stocking rates of >2 livestock units are sought. On the basis of the CF calculation, the N returned to the pasture is between 82% (N0) and 84% (N80) of the N intake. This led to field-N-balances between −119 kg N ha^−1^ year^−1^ (N0) and +706 kg N ha^−1^ year^−1^ (N80) with the most desirable field-N-balance obtained in the N20 treatment. The negative field-N-balance found in the N0 treatment implies an intensive N mining from soil, and consequently decreased pasture herbage yields can be expected in the long-term. This makes the efficiency of such a system questionable, unless other N sources are adopted, such as biological nitrogen fixation (BNF) from forage legumes introduced into the sward.

The results from the current study showed an improvement in the N-footprint as the amount of N-fertilizer was reduced. Another strategy for increased efficiency in relation to environmental impacts would be to increase the milk production per hectare [68]. However, milk production per hectare is closely related to stocking rate and is influenced positively by the nitrogen-use-efficiency (NUE).

The N-footprint is a widely adopted term used to indicate the uses and various losses of N within a system [56]. It is often based on the farm-N-balance or, as in the case of the current study, is based on predicted N-emissions. It indicates the N-pollution in relation to a given product (i.e., milk produced). The N-footprint in the current study ranged between 11 and 29 g N kg ECM^−1^ and was 62% reduced in the N0 treatment compared to N80 treatment. However, some doubt exists about the duration of this reduction at N0 due to expected yield losses in the long-term. The N20 treatment was ~50% reduced compared to the N80 treatment.

The imbalance of N in these pasture-based dairy systems should shift the focus on useful cycling of N to improve the NUE. This will create opportunities to try and reduce surplus inputs while reducing major N losses which would assist mitigation strategies [69]. The optimum NUE could be achieved by using revised N fertilizer rates as suggested by Viljoen et al. (2020) [8], which can further improve the CF of milk at farm level by decreasing N surpluses and thereby contributing to a lower N-footprint.

The NUE on pasture-based dairy farms can be further improved through management such as timing of fertilizer application, reducing the amount of fertilizer applied, and by incorporating forage legumes. The inclusion of legumes and forage herbs in pastures can reduce the impact on the environment of milk production from N-fertilized grassland [70,71]. The additional N supplied via BNF from forage legumes can also lower GHG emissions (reduced N_2_O compared with using N fertilizer) and reduce N-leaching, indicating more effective N-cycling in forage legume systems [72,73]. Legume forages allow N fertilizer requirements to be reduced and in turn reduce the N-status, which will enable increased farm N-efficiency [64] as well as reduced N-footprint. Organic farming systems have also followed this approach of combining livestock and crop production, and incorporating a high percentage of forage legumes in the crop rotations [72,73,74]. The botanical composition of pasture swards is affected by seasonal growth patterns of the constituent species. In the current study kikuyu was dominant in summer and ryegrass in winter, and the composition was comparable among the different treatments. However, the control plots (N0) had a higher component of unsown legumes (*Trifolium* spp.) compared to the other treatments and they contributed about 14% to the sward. Reinsch et al. (2020) [75] found at a comparable legume proportion in fertilized temperate permanent grassland with a BNF of 88 kg N ha^−1^ year^−1^, which partly explains the highly negative field N-balance in the N0 treatment. Nevertheless, the proportion of legumes and consequently BNF decreases in grass-legume swards when the rate of N fertilization is increased [8] and thus optimal fertilizer planning has to be adopted if this alternative N source should be utilized.

## 5. Conclusions

Applied mineral fertilizer (imported N onto the farm) contributed the largest fraction from the inputs category affecting the CF. The CF increased as the amount of added N fertilizer increased. An increased herbage yield was not observed when high rates of N fertilizer were applied and a low rate of mineral-N fertilization resulted in the lowest CF in this study. However, a level field-N-balance should be sought in order to sustain high yields and forage quality in the long-term. Fertilizer-N rates in excess of the N20 treatment used in this study are unlikely to provide a cost-effective response on grazed pastures and will only increase environmental impacts. The opportunity exists in pasture-based dairy farms to reduce further the environmental impact of milk production by optimizing efficiency as well as by management strategies to prevent over-fertilization or feeding excessive amounts of purchased concentrates. The contribution of irrigation played an important role in calculating the CF and the careful timing and application thereof could lead to a lower CF. This will ensure that pasture-based dairy farms can potentially mitigate the CF of milk in a profitable manner.

## Figures and Tables

**Figure 1 animals-11-01214-f001:**
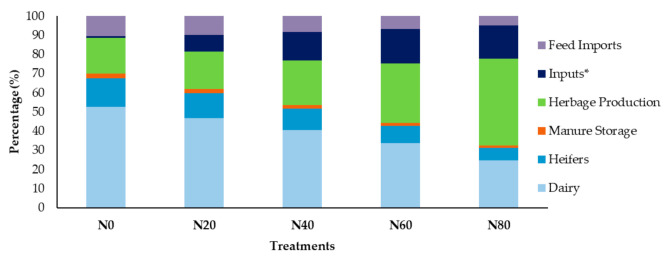
The relative contributions of the different system units to the product CF (%) among the different treatments (N0, N20, N40, N60 and N80). * Inputs (e.g., mineral fertilizer, lime, pesticides and seeds).

**Figure 2 animals-11-01214-f002:**
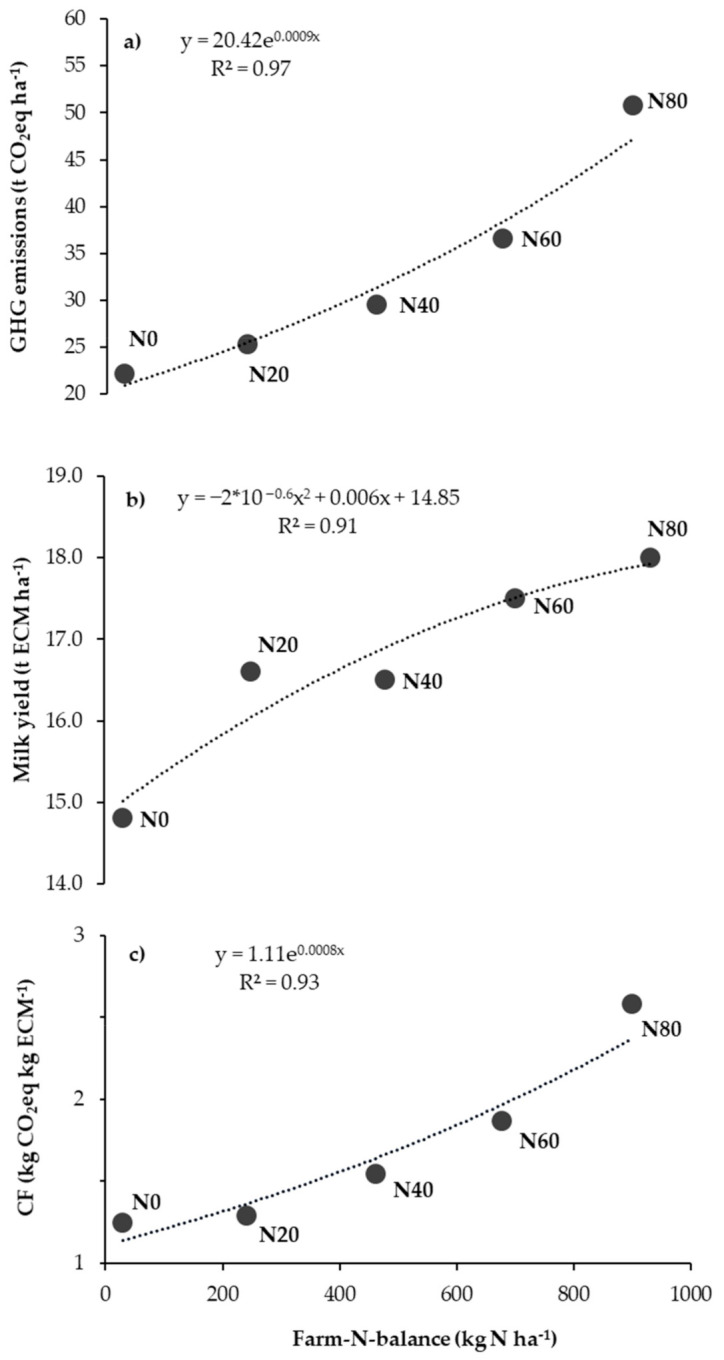
The relationship between (**a**) GHG Emissions (t CO_2_eq ha^−1^), (**b**) milk yield (t ECM ha^−1^) and (**c**) CF (kg CO_2_eq kg ECM^−1^) in relation to the farm-N-balance (kg N ha^-1^) between the different treatments (N0, N20, N40, N60 and N80). Soil carbon changes were considered. Means for the three experimental years are shown (n = 3).

**Table 1 animals-11-01214-t001:** Overview of the emission factors used for external resources. Adapted from [25].

Measurement	Emission Factor	Unit
Grass seeds	2.03	kg CO_2_eq kg^−1^
Ammonium-Nitrate as N	8.60	kg CO_2_eq kg^−1^
Tillage, rotary cultivator ^a^	75.42	kg CO_2_eq ha^−1^
Sowing ^a^	22.76	kg CO_2_eq ha^−1^
Fertilizing, by broadcaster	25.33	kg CO_2_eq ha^−1^
Mulching	21.24	kg CO_2_eq ha^−1^
Irrigation	0.43	kg CO_2_eq m^−3^
Milking	0.02	kg CO_2_eq kg^−1^
Shed operation	436.00	kg CO_2_eq LU^−1 b^
Slurry store and processing	0.06	kg CO_2_eq m^−3^

^a^ Grassland renovation was conducted every three years, ^b^ LU: Livestock unit.

**Table 2 animals-11-01214-t002:** Measured total herbage production (t DM ha^−1^), nitrogen yield (kg N ha^−1^) and energy yield (GJ NEL ha^−1^) contents of the pasture herbage. The averages were calculated between the three experimental years for the different N fertilization treatments used (N0, N20, N40, N60 and N80). SEM are shown in parenthesis.

Parameter	Year	Treatment				
		N0	N20	N40	N60	N80
Herbage Production (t DM ha^−1^)	1	19.3 ^Aa^ (9.3)	21.5 ^Aab^ (14.6)	21.1 ^Aab^ (9.0)	22.9 ^Ab^ (0.4)	22.6 ^Ab^ (0.5)
	2	17.4 ^Aa^ (4.7)	18.4 ^Ba^ (6.4)	18.9 ^Aa^ (0.3)	19.2 ^Ba^ (0.2)	20.5 ^Aa^ (0.4)
	3	18.7 ^Aa^ (11.4)	20.6 ^Aba^ (6.5)	20.3 ^Aa^ (0.4)	20.6 ^Aba^ (1.9)	21.5 ^Aa^ (0.7)
Nitrogen Yield (kg N ha^−1^)	1	519 ^Aa^ (26.8)	586 ^ABab^ (51.9)	657 ^ABb^ (34.4)	788 ^Ac^ (11.0)	849 ^Abc^ (29.0)
	2	511 ^Aa^ (24.6)	510 ^Aa^ (18.7)	557 ^Aa^ (16.6)	630 ^Bab^ (33.3)	749 ^Ab^ (9.7)
	3	582 ^Aa^ (38.9)	645 ^Ba^ (17.7)	707 ^Bab^ (13.5)	797 ^Abc^ (13.0)	900 ^Bc^ (35.3)
Energy Yield (GJ NEL ha^−1^)	1	123 ^Aa^ (5.7)	140 ^Aab^ (9.3)	137 ^Aab^ (6.5)	151 ^Ab^ (2.5)	148 ^Ab^ (2.9)
	2	112 ^Aa^ (2.7)	120 ^Bab^ (3.6)	123 ^Aab^ (2.1)	126 ^Bab^ (1.3)	135 ^Ab^ (3.1)
	3	119 ^Aa^ (7.5)	133 ^ABab^ (5.0)	132 ^Aab^ (3.0)	134 ^ABab^ (3.2)	141 ^Ab^ (4.9)

^AB^ no common capital letters indicate significant differences between years, ^abc^ no common lower-case letters indicate significant differences between treatments.

**Table 3 animals-11-01214-t003:** The potential average milk yields, GWP, CF and N-balance for the various treatments (N0, N20, N40, N60 and N80) over the three years. SEM are shown in brackets.

Parameter	Treatment				
	N0	N20	N40	N60	N80
Milk yield (t ECM ha^−1^)	14.8 ^a^ (0.7)	16.6 ^b^ (1.1)	16.5 ^c^ (0.9)	17.5 ^d^ (1.3)	18.0 ^e^ (0.9)
GWP (t CO_2_eq ha^−1^)	22.1 ^a^ (0.3)	25.3 ^b^ (0.4)	29.6 ^c^ (0.4)	36.6 ^d^ (0.1)	50.7 ^e^ (0.3)
GWP + soil carbon(t CO_2_eq ha^−1^)	18.4 ^a^ (0.2)	21.2 ^b^ (0.2)	25.4 ^c^ (0.2)	32.2 ^d^ (0.2)	46.2 ^e^ (0.5)
CF + soil carbon (kg CO_2_eq kg ECM^−1^)	1.3 ^a^ (0.1)	1.3 ^a^ (0.1)	1.6 ^b^ (0.1)	1.9 ^c^ (0.2)	2.6 ^d^ (0.2)
Farm-N-balance (kg N ha^−1^)	31 ^a^ (1.8)	241 ^b^ (3.8)	462 ^c^ (2.6)	677 ^d^ (5.0)	899 ^e^ (5.3)

^abcde^ no common lower-case letters indicate significant differences between treatments.

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
