# Peer review of "Environmental Impact of Rotationally Grazed Pastures at Different Management Intensities in South Africa"

_animals, 2021, doi:10.3390/ani11051214_

Round 1

Reviewer 1 Report

This article presents data on the farm-level N balance and carbon footprint in a grazing/overseeding system in South Africa under a range of N input treatments. By European standards, the N use is colossal and highly inefficient - the authors themselves admit that most of the N treatments are well over normal management regimes and the local fertilizer guidelines, which begs the question of why such high N treatments were used in the study (better justification should be provided). If the idea is to move towards more sustainable management, treatments <400 kg N/ha should have been privileged.

In general, this paper represents a case-study with clear local interest.  It is not written targeting a broad international audience, and although the reference list is extensive, it does not cite key, influential papers in the field of farm-level N balance and carbon footprints, some of the sources of the references cited are unconventional/ borderline scientific and many others are clearly not the most appropriate or relevant papers. (The references need to be checked and improved). This aside, the study appears to have been well conducted, the calculations methodology is based on published protocols, and the statistical approach is generally fine (although less emphasis should be given to the average data eg Tables 4 & 5 for which there are no error bars/statistics, and which could lead to misleading interpretations). Finally, the introduction is poorly written (lacks structure and logic of ideas, hypotheses need improvement), and the discussion is overlong and overly-repetitive of results. The overall impression is that the results are largely confirmatory (the authors should better highlight and focus on the novel aspects of their work). Suggestions for improvements are below:

Summary: Delete L20-21. Specify farm-scale N balance. Check grammar and tenses. Rewrite L26-28 more clearly.

Abstract: Delete L36. Rewrite L37-41. What is a levelled field N balance?

Introduction: Rewrite L47-49, and L64-65 (not good English, unclear). L73: replace observed by recorded. Delete L83-89.

Reorganise L90-105: Start with L90-94, then 100-102, then 95-96 (put refs at end of the sentence). End with “These studies suggest that the CF of milk ranges from 0.94-2.07 at the farm gate, but little is known on the impacts of N inputs on the CF of milk in S.African dairy systems”. Delete 103-105.

Delete H1 and H3 (neither are an acceptable hypothesis, or worded correctly).

Methods: Delete 132-135 and section 2.2 (unnecessary). Move section 2.4 before 2.3, refer to grazing events or cycles throughout, and delete 169-174 (this is results and should be mentioned later). L186: reword as ‘used to take biomass samples at a height of 3cm above soil level’. Reword L210 (production circumstances??). L217: database??

Results: move Table 4 & 5 to appendix- focus instead on Table 6 (adjust results and discussion text accordingly).

Discussion: L375: provide intensive N cycling?? Rewrite L376-382 (not clear), and avoid repeating this point later on. L390-400 is either clumsily written or does not make sense – rewrite. Delete L419-437 and L470-476. Reduce L552-559.

Reviewer 2 Report

Overall: The study is definitely interesting and suitable for publication. However, some improvements would be desirable: The section Material and Methods is a bit confusing. Possibly one could create a better overview here by tables or diagrams. In addition, it is not quite clear to me to what extent the production of the N-fertilizer as well as the concentrated feed have been included in the calculations? This should be described very clearly, because the choice of the system boundaries has a strong impact on the results (should also be taken up in the discussion). Also a separate section "limitations of the study" would possibly be helpful, since in principle many assumptions are made when calculating the CF.

Line 63-64: “However, this can affect the stocking rate…” How does it affect stocking rate?

Line 67-68: “An increase in stocking rate and milk production are associated with pasture under irrigation in combination with high fertilizer usage” rephrase

Line 76-77: “and other forms of environmental pollution” name them.

Line 83-85: “For this reason, increasing urgency to focus on adapting current agricultural production systems, to achieve a sustainable balance between profitability and the environmental impact effects, of dairy farming exist” Rephrase

Line 107-110: “Against this background, this article aims to evaluate dairy-pasture systems in terms of different N fertilization levels on herbage yields as well as calculating the associated CF of milk using the measured parameters. The present paper endeavours to fill the gap in the literature while contributing to sustainability by assessing the CF of milk produced at different management intensities in pasture-based systems in South Africa” redundant

125-128: “The area has a temperate climate with mean monthly temperatures ranging between 7-18 °C and 15-25 °C in winter and summer, respectively. Rainfall is evenly distributed throughout the year and has a mean annual average precipitation of 661 mm (ten year data)” reference?

134-135: “where they also receive some concentrate feed in a herringbone layout milking parlor” Which type of concentrate?

140: “Error! Reference source not found..” ???

161: Table 1: Please use a single line for each year (2/3)

Line 172-174: “However, the control plots (N0) had a higher component of unsown legumes (Trifolium spp.) compared to the other treatments and they contributed 14% to  the sward. It is known that the contribution of legumes in a sward increases as the added  amount of fertilizer decreases” This should be a part of the discussion.

250: Co2 equivalents: you used this term before

252: ECM: same

259: Error… ??

278: Error..?? References should not be part of the result section.

Figure 2: What “Inputs”? Should be defined.

347, 349, 350: error…

377: “Concentrates are fed between 4-6 kg cow-1 day-1 in the specific production area where the study was conducted.” – in terms of sustainability it would be interesting where these concentrates are imported from. It should also be mentioned, that the production of these concentrates might take place under high amount of N fertilization, too. For a detailed CF of concentrate production look at

439: error..

532: This led to field-N-balances between -119 kg N ha-1 year-1 (N0) and +706 kg N ha-1 year-1 (N80) with the most desirable field-N-balance in the N20 treatment” theoretically yes. However, you didn’t do soil samples so this remains an assumption. Especially since, you reported in the material and methods section that the N0 treatment has a higher proportion of legumes. The possibility to control the N budget via legumes is generally missing in the discussion.

588: “Application of low rates of N fertilizer will not be sustainable in the long-term as soil nutrients would be depleted leading to soil mining” we don’t know that clear enough to make such a statement, I think. This is contradicted by the fact that no N-fertilizers are used in organic farming and their soil contents are not necessarily lower than in conventional farming if they are managed correctly.

Round 2

Reviewer 1 Report

The revised manuscript (provided by the section editor) is significantly improved. Some minor revisions are required (see below):

L15: to which

L17: dairy system environmental efficiency

L36: appears sufficient

L48: “bought in” forage

L64: circulating within, or being lost from,

L68: pastures in some areas are additionally…

L81: which allow

L99: delete ‘using the measured parameters’

L149: specify the number of splits per year. If I understand the text correctly, there are 12 grazing events and hence 12 splits.

L176: excessive precision for the ring area. Provide the diameter of the rings instead.

L200 & L203: delete ‘fed’

L239: replace ‘ratio’ by ‘fraction’

L321: What does ‘avoiding the deduction of gaseous N losses’ mean? Reword sentence more clearly.

L353: …due to within-field N cycling and high N returned to…

L362: Replace ‘leading to’ by ‘likely promoting’

L365: of N input

L366: local advisory information

L373: comparable among treatments except…

Delete L375-376. Start L377 with ‘As expected, the treatments…’

L412: regionally-specific

Delete L416-419 ‘Therefore…’ as this contradicts statements made later on.

L499: Replace ‘indicates’ by ‘implies’

L533: …in turn reduce the N status…

L551-553: Delete ‘high N rates lead to negative environmental effects’ and combine the two sentences in one.

L556: Replace ‘worthwhile’ by ‘cost-effective’

Reviewer 2 Report

All my previous comments and criticism has been edited and improved. This paper is acceptable.

Author Response

The authors wish to thank the anonymous reviewer for their inputs in improving the manuscript.